# Optimization of Transspinal Stimulation Applications for Motor Recovery after Spinal Cord Injury: Scoping Review

**DOI:** 10.3390/jcm12030854

**Published:** 2023-01-20

**Authors:** Muhammad Uzair Rehman, Dustin Sneed, Tommy W. Sutor, Helen Hoenig, Ashraf S. Gorgey

**Affiliations:** 1Spinal Cord Injury and Disorders, Hunter Holmes McGuire VA Medical Center, Richmond, VA 23249, USA; 2Department of Biomedical Engineering, Virginia Commonwealth University, Richmond, VA 23284, USA; 3Department of Physical Medicine and Rehabilitation, School of Medicine, Virginia Commonwealth University, Richmond, VA 23284, USA; 4Physical Medicine & Rehabilitation Service, Durham VA Health Care System, Durham, NC 27705, USA; 5Geriatrics Division, Department of Medicine, Duke University, Durham, NC 27710, USA

**Keywords:** transspinal stimulation, transcutaneous, motor outcomes, spinal cord

## Abstract

Spinal cord injury (SCI) is a debilitating condition that can significantly affect an individual’s life, causing paralysis, autonomic dysreflexia, and chronic pain. Transspinal stimulation (TSS) is a non-invasive form of neuromodulation that activates the underlying neural circuitries of the spinal cord. Application of TSS can be performed through multiple stimulation protocols, which may vary in the electrodes’ size or position as well as stimulation parameters, and which may influence the response of motor functions to the stimulation. Due to the novelty of TSS, it is beneficial to summarize the available evidence to identify the range of parameters that may provide the best outcomes for motor response. The PubMed and Google Scholar databases were searched for studies examining the effects of TSS on limb motor function. A literature search yielded 34 studies for analysis, in which electrode placement and stimulation parameters varied considerably. The stimulation protocols from each study and their impact on limb motor function were summarized. Electrode placement was variable based on the targeted limb. Studies for the upper limbs targeted the cervical enlargement with anatomical placement of the cathode over the cervical vertebral region. In lower-limb studies, the cathode(s) were placed over the thoracic and lumbar vertebral regions, to target the lumbar enlargement. The effects of carrier frequency were inconclusive across the studies. Multisite cathodal placements yielded favorable motor response results compared to single-site placement. This review briefly summarized the current mechanistic evidence of the effect of TSS on motor response after SCI. Our findings indicate that optimization of stimulation parameters will require future randomized controlled studies to independently assess the effects of different stimulation parameters under controlled circumstances.

## 1. Introduction

Spinal cord injury (SCI) is a highly debilitating condition, with nearly 20,000 new cases per year [1]. SCI can have a wide range of motor effects, ranging from nearly full recovery to permanent, severe disability. SCI can be traumatic or non-traumatic, resulting in changes in spinal nerves’ connectivity and signal transmission. Paralysis or paresis of the lower extremities and trunk is classified as paraplegia/paresis, whereas paralysis or paresis of the upper extremities, trunk, and lower extremities combined is called tetraplegia [2]. A clinical diagnosis of “complete” SCI indicates no preserved motor or sensory function below the lesion, whereas an “incomplete” SCI may have varying degrees of preserved sensory and motor function [3]. In the United States, 47.2% of all SCIs result in incomplete tetraplegia, 19.6% in incomplete paraplegia, 20.2% in complete paraplegia, and 12.3% in complete tetraplegia, while 0.7% result in complete recovery of sensory and motor function [1]. The debilitating effects of SCI have led to researchers exploring different treatments over the years. A key area of interest has been the use of various techniques to target neuroplasticity via neuromodulation approaches. The International Neuromodulation Society describes neuromodulation as “a field of science, medicine, and bioengineering that encompasses implantable and non-implantable technologies, electrical or chemical, for the purpose of improving the quality of life and functioning of humans.” [4] Neuromodulation has many applications, including managing chronic pain, spasticity, and loss of physical function. One method for neuromodulation is the use of electrical stimuli [4]. Electrical stimuli can be delivered by several methods, including transcutaneous electrical neural stimulation, neuromuscular electrical stimulation, functional electrical stimulation, and transspinal stimulation (TSS) [5]. Some of the most exciting developments in neuromodulation involve the use TSS for the treatment of persons with SCI.

TSS is a non-invasive form of neuromodulation. TSS as a modality can be split into two subtypes: direct-current TSS and pulsed TSS [6]. This review will focus on pulsed TSS, a distinction which should be noted, despite the use of the abbreviation TSS throughout the manuscript. TSS involves the placement of stimulation electrodes over the skin, with the cathodes between interspinous processes and the anodes on the opposing bony landmarks such as clavicles, tips of the shoulders, lower abdominal region, or iliac crests to induce stimulation of spinal cord neural circuitries [7,8].

A growing body of evidence supports the idea that TSS promotes functional recovery in humans with SCI [7,8,9,10]. Motor and sensory improvements were observed when TSS was applied with [8,11] or without [9] other therapies such as buspirone [8], exoskeleton training [11], or task-specific activities focused on hand strength training to increase grip strength and function of the hand [12]. TSS applied at the cervical level can improve upper-extremity function in people with tetraplegia [7]. After a single session of TSS, Benavides et al. found that TSS had an excitatory effect at the spinal level, which was measured by cervico-medullary evoked potentials, and an inhibitory effect at the cortical level as measured by intracortical inhibition [7]. These changes were associated with improved upper-extremity function in individuals with tetraplegia. An assessment of spinal cord evoked potentials demonstrated that these results were due to increased spinal network excitability with tonic TSS. Gad et al. proposed using cervical TSS twice weekly for four weeks with participants in an upright sitting position as a non-invasive approach to modulate the cervical network segments [12]. The training session lasted for 1–2 h and involved stimulation parameters that enabled motor control rather than inducing motor function. Maximum voluntary hand grip forces increased by 325% in the presence of stimulation and 225% when grip strength was evaluated without simultaneous stimulation in participants with chronic tetraplegia 1–21 years post-injury [12]. Maximum evoked responses quantified by EMG amplitude in both the flexor digitorum and extensor digitorum also increased significantly across the studied cohort. Subjects demonstrated improved upper-extremity function starting from the first training session, as demonstrated by their abilities to generate a greater maximum voluntary hand grip force [12].

The protocols included in this review varied based on parameters and outcome measures, such as the utilization of the posterior root muscle (PRM) reflex [13] to establish a cut-off for stimulation amplitude. The PRM is a short-latency spinal reflex resulting from TSS; it occurs due to the activation of proprioceptive fibers in the posterior nerve roots, which activate motor neurons in the anterior aspect of the spinal cord. The definition of the motor threshold varies considerably among these studies. For example, Shapkova et al. defined the motor threshold as the amplitude required to elicit motor evoked potential (MEP) on an EMG in at least four lower-extremity muscles (bilateral rectus femoris, biceps femoris, gastrocnemius lateralis, and tibialis anterior) [14]. Both Wu et al. and Kumru et al. defined the motor threshold as the amplitude required to elicit an MEP of 50 microvolts in the adductor pollicis brevis in 5/10 repetitions [15,16]. In contrast, Al’Joboori et al. defined the motor threshold as the amplitude that elicited visible muscle contractions. Other studies did not specify their protocol for determining the exact motor threshold [17].

The current review aims to summarize the stimulation protocols and electrode parameters of TSS studies that were directed toward enhancing upper- and lower-extremity motor function. We have also included studies on healthy able-bodied controls to facilitate understanding of how TSS was managed differently in persons with SCI. We are hopeful that the review will offer a clear description of the included studies, highlighting the protocols utilized and providing a better understanding of the parameters of TSS. It will also highlight the mechanisms which underlie TSS-enabled motor function [18,19,20,21].

## 2. Methodology

The PubMed and Google Scholar databases were searched from December 2021 to July 2022 with the query (“transcutaneous” OR “non-invasive” OR “Transspinal” OR “trans-spinal” OR “Transpinal”) AND “stimulation” AND “spinal cord” AND “spinal cord injury” AND “upper limb” AND “lower limb.” All titles were read; those that had “Transspinal,” “paralysis,” “stepping,” “lower limb,” “upper limb,” “hand function,” and “SCI” were included for abstract reading. Studies were included for full reading if they had motor or neurophysiologic outcomes based on EMG. Review articles [22] were excluded from full reading; however, their references were examined for relevant studies. This search identified 12 studies on upper-extremity function and 22 on lower-extremity function. All included studies were read thoroughly, and their protocols were summarized. In the following sections, we will outline the information on electrode placement, stimulation amplitude, waveform, pulse width, frequency, and carrier frequency. This review will also look at study outcomes and the simulation parameters that were used to achieve them.

### 2.1. Mechanism of Action

Several studies have explored the mechanisms that underlie TSS-enabled motor function, which can be summarized as follows. The effects of TSS on motor control can be attributed to either supraspinal or spinal mechanisms. In SCI, the corticospinal tract is an important target for the recovery of motor functions. Inducing voluntary motor outcomes depends on the effectiveness of the connections between the corticospinal axons and motor neurons in the spine. Current neurophysiologic data indicates that TSS facilitates the formation of new synaptic connections between spinal interneurons and motor neurons via the stimulation of dorsal afferents, thus allowing restoration of supraspinal control of motor function.

In the spinal region TSS activates the neural circuitry by recruiting afferent fibers located in the posterior root and elevating spinal circuit excitability [18,19]. Gerasimenko et al. identified that the excitability of spinal interneuronal networks could be modulated without directly resulting in action potential production [20]. TSS is known to recruit afferent group Ia and group II fibers in the posterior root while also engaging the motor neurons in the anterior horn; these findings can be viewed by carefully studying evoked motor potentials [20,21]. Furthermore, as the stimulation amplitude rises, there is an increase in the types of afferent fibers (Ib, group II muscle spindle, and cutaneous afferents) and interneurons that are recruited, which results in motor neurons and interneurons achieving a base state closer to the firing threshold and become more responsive to the descending signals in the injured spinal cord [20].

The supraspinal effect of TSS may result from activation of the axons of the extrapyramidal tract in the subcortical white matter. TSS delivered at submotor thresholds allows electrical stimuli to be delivered to supraspinal and spinal centers to induce mechanical events. The arrival of the presynaptic signal to the corticospinal tract prior to the activation of the motor neurons causes a discharge in the corticospinal transmission. This may result in neural plasticity of the corticospinal and motor neuron synapses that can facilitate motor function in SCI, and modulation of the remaining synapses can target and enhance voluntary motor control in persons with SCI [23].

Stimulation of the dorsal root is not the only route through which TSS may modulate the underlying neural circuitries [24]. In fact, cutaneous activation occurs over a range of stimulation amplitudes that are lower than the motor threshold required to induce activation of the Ia fibers. Electrical stimulation over the spinal cord has been shown to cause inhibition via interneurons in the laminae I–III, making it important to consider that the inhibitory neurons play a role in the beneficial effects of TSS by enhancing the dorsal GABAergic systems [24]. Apart from the depolarization of the sensory afferents in the dorsal root and the dorsal horn to transsynaptically recruit motor pools, the polysynaptic connections of the mechanoreceptors in the skin are believed to act on the sensory and motor pools in the spinal cord; this occurs through the connections that these interneurons establish between the spinal levels higher and lower than the point of injury [24].

### 2.2. Electrode Placements

Proper anatomical placement of electrodes is vital to target the correct spinal regions and provide the ideal location for interfacing with the spinal network, as noted in Table 1a,b. There are two types of electrodes: the cathode, or negative electrode, and the anode, or positive electrode. In the studies reviewed, the cathodes were either placed at a single anatomical location (referred to as single-site stimulation) or at more than one anatomical location (referred to as multisite stimulation). Several studies were based on the idea that targeting the cervical and lumbar enlargements would facilitate the stimulation of a large number of neural circuits associated with the activation of muscles in the respective limbs.

The lower-limb studies with able-bodied participants that adopted a single-site arrangement used cathode placement between spinous processes T11/T12 [29,41], L1/L2 [41], or bilaterally on each side of the spinous processes at L1/L2 [40]. Studies using a multisite placement opted for midline spinous processes of T10 and L1 [34], midline at C5, T11, and/or L1 spinous processes [35], 8 cm caudal and 4 cm rostral around the interspinous space T11–12 [30], and between the spinous processes of T10–T11, T11–T12, and T12–L1 midline [36]. The anode in the studies with able-bodied participants was placed bilaterally over the iliac crest, except for Manson et al., who placed it over the abdomen centrally [40] and placed the cathodes bilaterally in between L1/L2.

Single-site studies for lower limbs which enrolled participants with SCI placed the electrodes at the midline between T10/T11, T10/L1 [32,38], the midline over T12, the midline between the T11/T12 [10,13,31] spinous processes, or para-spinally at T11/T12 [13]. Another study placed the cathode between T12–L1 [17]. In contrast, multisite studies placed cathodes at the midline between spinous processes T11–T12 or over Co1 [12], between spinous processes T11–T12 [39] or L1–L2 [44], from T10/T11 to L4/L5 [33], and midline between spinous processes T11–T12 or over the coccyx [37]. A study by Samejima et al. applied cathodes at four different anatomical sites over the midline at C3/C4, C6/C7, T11, and L1 [43]. In the studies where participants had SCI, the anodes were placed over the lower abdomen or bilaterally over the iliac crest.

In upper-limb studies with able-bodied participants, single-site cathode placement was over C6, C7, or T1 [27,28], while one study utilized an electrode that ranged from C7–T1 [18]. A multisite study applied a cathode each at midline over C3–C4, C6–C7, and over the T1 and L1 spinous processes [21]. The anodes were applied over the iliac crests, midline over the anterior neck, and bilaterally over the clavicles.

Single-site upper-extremity studies with participants who had SCI opted for cathode placements over the C5 spinous process [8], midline between the C5–C6 spinous process [7], and over the midline 4 cm caudal to the C7 spinous process [15] in a longitudinal arrangement. The studies that utilized multisite placement of the cathode among persons with SCI positioned one at the midline over the C3–C4 spinous processes and another midline at the C6–C7 spinous processes [9,12]. Another study positioned the cathode at the midline above and below the injury level [26], while one study used a cathode that covered C5–T2 [25]. The anodes were placed over both clavicles or iliac crests, and one study applied them 2 cm over the sternal notch.

#### Electrode Configuration

Another factor associated with the electrodes is their shape. Several studies applied rectangular [10,11,19,32,38] and/or circular [11,19,31] electrodes for the cathode, while rectangular-shaped electrodes were chosen as anodes. Though no study specifically identified the reasoning behind their selection, it was observed that in multisite cathodal placements, a circular electrode was placed at a higher vertebral level as compared to pair of rectangular electrodes for lower vertebral locations [11,31]. Manson et al. employed an electrode arrangement in the form of a grid [40].

Sayenko et al., conducting a study with able-bodied participants, opted to place the electrodes between the spinous processes of T10/T11, T11/T12, and T12/L1 [19]. They observed that the position of the cathodes changed the intensity of the response in muscles, such that stimulation at T10/T11 caused the vastus lateralis and rectus femoris to respond with a higher magnitude as compared to the medial hamstring and soleus muscles. On the contrary, when stimulation was delivered at T12/L1, the medial hamstring and the soleus showed a greater amplitude response.

Sasaki et al. sought to determine what cathode placement would best facilitate the PRM reflex, which is a short-latency spinal reflex that can be evoked with TSS [27]. De Freitas et al. assessed the effects of different cathode–anode arrangements, observing that distal hand musculature was most responsive to an orientation with the cathode over T1 and the anode over the anterior neck [28]. Gad et al. compared the effects on maximum grip strength facilitated by multisite (C3/C4 and C6/C7) vs. single-site (C3/C4 or C6/C7) stimulation in participants with SCI [12]. The study’s outcomes reported that distal muscles displayed more forceful voluntary contractions in response to multisite stimulation. Similarly, Gerasimenko et al. compared the effects of multisite (C5/C6, T11/T12, and L1/L2) vs. single-site (C5/C6 or T11/T12 or L1/L2) stimulation in able-bodied individuals while in a lying position [35]. It was observed that stimulation at T11 resulted in motor evoked potential (MEP) of lesser magnitude than stimulation at the other three sites simultaneously in both the medial gastrocnemius and bicep femoris muscles. Wu et al. evaluated the effects of several electrode arrangements in both able-bodied participants and participants with SCI [15]. They observed that compared to cathode-anterior arrangements at C4/C5 with biphasic waveforms, cathode-posterior arrangements at T2/T4 with biphasic waveform orientations elicited large muscle responses in the abductor pollicis brevis and abductor digiti minimi at lower intensities. Furthermore, De Freitas et al. conducted two experiments in supine position: one to optimize cathode placement and one to optimize anode placement [28]. In phase one, upper-extremity MEP was measured. The cathode was moved from the spinous processes of C6 to C7 and from C7 to T1 with the anode fixed on the anterior neck. The optimal cathode location was chosen and held constant for each participant. In contrast, the anode location varied among one anode on the anterior neck or two anodes on bilateral clavicles, two anodes on the iliac crests, and one anode 4 cm below the cathode on the posterior neck. Out of the four positional arrangements, placing the cathode at T1 yielded the most MEP in the hand musculature.

Future studies should aim to further explore the role of the anatomical position of the cathode by developing protocols that compare outcomes between different multisite vs. single-site stimulation studies. Although not widely investigated in the current literature, the body’s positioning could affect the area targeted by the stimulation and should be looked into in further detail in the future.

### 2.3. Waveform

Studies included in this review reported using two different waveforms, biphasic or monophasic. Among the lower-limb studies with able-bodied participants, four reported using a biphasic waveform [29,30,35,40], whereas another four reported using a monophasic waveform [34,36,37,41]. Among SCI studies, five studies reported the use of a biphasic waveform [10,31,42,43,44], five reported the use of a monophasic waveform [13,14,17,19,33], and another three [11,32,38] studies did not report what type of waveform was delivered.

Similarly to the lower-limb studies mentioned above, those focused on the upper limbs also applied either a biphasic or monophasic waveform. In studies with able-bodied participants, two applied a biphasic waveform [16,21], three applied a monophasic waveform [18,27,28], and one study examined both mono- and biphasic waveforms [15]. Among seven studies that included SCI to enhance upper-extremity function, two studies used exclusively biphasic stimulation [7,9], one used exclusively monophasic stimulation [25], three used a combination of monophasic and biphasic stimulation [12,15,26], and one did not report the type of waveform used [8].

The comparison between the two different types of waveforms has not been studied in great depth, and most study protocols do not provide a rationale for their selection. The selection of a biphasic waveform may be attributed to the fact that it does not have an electrochemical polarization effect, hypothetically improving tolerance and application at a higher stimulus amplitude [15] and possibly protecting against tissue damage, especially below the level of the injury. In contrast, a monophasic waveform causes depolarization of the neural membrane by generating negative charges from the cathode [45]. One study that examined the effect of monophasic vs. biphasic waveforms [26] reported that biphasic waveforms facilitated fine motor control [26]. In contrast, monophasic waveforms facilitated the performance of strength-related activities [26]. Additional studies are needed to examine the effects of different waveforms on motor response. The aforementioned evidence may support the use of a biphasic waveform for application of TSS.

### 2.4. Stimulus Amplitude

Stimulus amplitude is a parameter that varied with each study, as summarized in Table 2. Additionally, motor responses may vary based on the amplitude of the stimulation. Some studies quantified stimulus amplitude in terms of current (mA), while three studies by Hofstoetter et al. reported amplitude in terms of voltage (v) [10,31,42]. It is useful to review studies of stimulation parameters from two perspectives: (a) classified by upper- vs. lower-limb stimulation, subdivided by studies in able-bodied persons vs. persons with SCI; and (b) classified according to stimulation amplitude.

Stimulation amplitude adjusted to submotor, motor, or supramotor thresholds affected the examined outcomes. A submotor threshold amplitude was used in several of the reviewed studies [10,17,31]. Hofstoetter et al., noted that a submotor threshold amplitude resulted in a greater gain in the range of motion of ankle dorsiflexion and a decrease in the clonus activity of the tibialis anterior and calf muscles compared to no stimulation [31]. Another study reported that at a submotor threshold, while in standing position, the stride length was increased from 1.13 m to 1.32 m and facilitated a more fluid multi-joint movement cycle [10]. Inanici et al. reported that using a submotor threshold stimulus significantly increased participants’ pinch force and their graded redefined assessment of strength sensibility and prehension [26]. The use of a 100% motor threshold amplitude was applied by Gad et al., and they observed an increase in grip strength. Another study noted increased step-like movement, a decrease in the time required to perform a motor function, and increased MEP as compared to no stimulation [12]. Studies that used a supramotor threshold [14,15,16] indicated increases in the walk time while using an exoskeleton [14], decreased response latency [15], and augmented hand muscle strength [16] in a sitting position. Few studies compared the effects of varying the stimulation amplitudes. Only three studies examined amplitudes that ranged from the submotor threshold to the supramotor threshold. Kumru et al. studied the effects of different stimulation intensities on hand motor function and strength in the upper limbs of able-bodied participants [16]. The study utilized subthreshold levels of 80% and 90% of the resting spinal motor threshold along with a supramotor threshold value of 110%. It reported that using 90% of the amplitude of the spinal motor threshold induced better results than other stimulation intensities [16]. Wu et al. measured the motor threshold and then delivered stimulation that ranged from 80% to 200% of this level to participants in a seated position. The study reported that the higher stimulation amplitudes resulted in shorter latency periods by up to 2.5 ms in the abductor policis brevis muscle [15].

In lower-limb studies with able-bodied participants, the stimulus amplitude was either based on evoking a stepping-like motor response [29,35,41] or the highest tolerable amplitude [30,34,35,36,40]. However, some studies had a maximum safety threshold they would not exceed [29,30,35]. In lower-limb studies with SCI participants, researchers chose an amplitude that resulted in the PRM reflex [13], producing the desired outcome of stepping [11] while in a supine position, eliciting MEP [33], or producing paresthesia [10,17,31]. These studies aimed to stimulate at a submotor threshold level rather than inducing motor function. Several studies did not list the reason behind the selected amplitudes [10,19,32,38].

In upper-limb studies with able-bodied participants, the protocol was different in each study. Three studies tested a range of intensities from the subthreshold to the suprathreshold [15,16,28], another adopted the maximum tolerated amplitude [21], and one study chose the amplitude at which the participant reported paresthesia [27]. In persons with SCI, amplitudes that maximized the motor response [7,8,12] or protocols that tested stimulation amplitudes ranging from the submotor threshold to the supramotor threshold of the resting motor potential [15,25,26].

McHugh et al. used stimulation amplitudes either at the highest tolerated level or at the submotor threshold to activate the lower-limb muscles in participants with SCI. The study used the 10 m walk test (10MWT), 6 min walk test (6MWT), timed up-and-go test, and walking index for spinal cord injury II as primary outcome measures. The results showed that all participants gained significant gait speed, increased endurance, and improvements in functional mobility [39].

In sum, the selection of stimulation amplitude varied considerably across studies, rendering it difficult to determine which method results in the greatest functional benefit and for whom. This level of variability also makes replication difficult, and it is not as informative scientifically (i.e., which method for stimulus selection is optimal). On the other hand, the generalization of a protocol which specifies a base amplitude would not be ideal, as everyone has a different tolerance level. In this case, the method of ascertaining tolerance needs to be defined and implemented consistently. This is an important question to be addressed by the field.

### 2.5. Pulse Width

Pulse width is another factor necessary for the deliverance of a stimulus. Among the lower-limb studies with able-bodied participants, five studies reported the use of a 1 ms pulse width [34,35,36,40,41], two studies reported a pulse width of 0.5 ms [29,37], and one study applied a 2 ms pulse width [30]. Among SCI studies, eight studies reported the use of a 1 ms pulse width [13,17,19,33,39,42,43,44], and two reported a 2 ms pulse [10,31] width; three studies did not report the used pulse width [11,32,38]

Three reported using a 2 ms pulse width in studies that targeted the upper limbs in able-bodied participants [18,27,28], and two used a 1 ms pulse width [16,21]. Similarly, in SCI studies, one study reported the use of 0.2 ms pulse width [7], while five studies used a 1 ms pulse width [9,12,15,25,26], one used a 2 ms pulse width [15], and another did not specify the used pulse width [8].

Wu et al. observed that in able-bodied participants a 1 ms pulse caused lower discomfort levels as compared to a 2 ms pulse width with mono- and biphasic waveforms, although the strength of the stimuli was not mentioned [15].

The use of different pulse widths may alter the recruitment of neurons; it has been observed that a shorter pulse width requires a higher amplitude to cause neuron activation, in contrast a longer pulse width, which can cause activation of neurons at lower amplitudes [46]. Furthermore, it is important to identify the pulse width that best facilitates a decrease in pain and improves motor function. This may explain why several of the reviewed studies have recommended the use of a 1 ms pulse width.

### 2.6. Frequency

One stimulus-related parameter in the application of TSS is the frequency, as summarized in Table 3. Among the lower-limb studies with able-bodied participants, four studies used a 30 Hz [29,34,37,41] frequency, and one study reported the use of a 5 Hz frequency [35], while another two did not report the frequency of the stimuli [30,36]. Gorodnichev et al. applied several different frequencies (1, 5, 10, 20,30, and 40 Hz) [29] which resulted in inducing involuntary step-like movement in participants. Other studies have applied different stimulation frequencies, such as 30 Hz at T11 accompanied with 5 Hz at the coccyx [37], and another study used 30 Hz at T11–T12 accompanied with 0.3 Hz at L1 [41]. In SCI studies, stimulation was applied at 30 Hz [10,11,13,17,19,33,38,43] in eight studies, while three reported using 50 Hz [31,38,39], and one reported using 20 Hz [32]. To observe the role of frequency, Kaur et al. used multiple frequencies (30, 50, 70, and 90 Hz) [38] delivered at a single stimulation site and reported that the application of higher frequencies (70 Hz and 90 Hz) caused better activation of the quadriceps as compared to lower frequencies (30 Hz and 50 Hz). Shapkova et al. applied frequencies of 1 Hz, 3Hz, and 67 Hz [14] and reported that the application of 67 Hz resulted in decreased spasticity and increased exoskeleton steps. Sayenko et al. applied 5 Hz, 15 Hz, 25 Hz, and 30 Hz [19], and they observed that 15 Hz provided the most robust effects on standing in all participants, while 25 Hz caused the lowest muscle amplitudes while facilitating standing. In comparison, one multisite study by Gad et al. that applied frequencies of 30 Hz at T11 and 5 Hz at Co1 [11] reported a decrease in the mean step cycle from 2.13 s to 2.03 s.

In the upper-limb studies with able-bodied participants, the stimulus was set at a frequency of either 30 Hz [16,21,27], 0.2 Hz [15], or two pulses with a 50 ms interval in between [28,40]. In comparison, those studies with SCI participants used a frequency of 30 Hz [7,8,9,12,26], except for Manson et al., in which burst stimuli were delivered at a frequency of 0.2 Hz or continuously at 30 Hz [40] with similar motor threshold levels.

In addition to the role of stimulus frequency on the motor outcome, some studies examined the effects of the frequency on the participants’ spasticity. Hofstoetter et al., showed that a frequency of 50 Hz resulted in decreasing spasticity, exaggerated reflexes, and improvement in passive movements [31]. Shapkova et al. also reported a beneficial outcome on spasticity with their TSS protocol; they utilized three different frequencies (1 Hz, 3 Hz, and 67 Hz), observing that a higher frequency facilitated lowering spasticity and enabled exoskeleton-assisted walking [14]. Al’joboori et al. also applied a high stimulus frequency of 30 Hz to counter the effects of spasticity. The study reported improvements in lower-limb voluntary motor control under the protocol utilized [17].

### 2.7. Carrier Frequency

Carrier frequency as a stimulation parameter was not utilized by all the studies that were reviewed, as summarized in Table 3. Of the lower-limb studies with able-bodied participants, five studies reported using a carrier frequency, with two applying a frequency of 5 kHz [40,41] and three applying a frequency of 10 kHz [29,35,37]. Among studies with SCI participants, five reported using a frequency of 2.5 kHz [32,38] or 10 kHz [19,43,44]. In the studies of the upper limbs in able-bodied participants, three reported using a carrier frequency of 10 kHz [16,21,27]. However, among studies where participants had SCI, three studies applied 10 kHz [9,12,26], and one study used 5 KHz [7].

The role that carrier frequency plays is not fully understood. Gerasimenko et al., Inanici et al., and Gorodnichev et al. reported that using a carrier frequency may lead to improved muscle strength and mitigate discomfort [9,29,35]. Manson et al. provides insight into the claim that carrier frequency reduces the patient’s discomfort [40]. This study found that participants tolerated a significantly higher stimulation amplitude when a carrier frequency was applied, with subjects tolerating an amplitude more than twice as high when a carrier frequency was applied vs. when there was no carrier frequency (582 mA vs. 260 mA) [40]. However, the motor threshold was also increased by roughly the same factor (195 mA vs. 70 mA) with the application of a carrier frequency. There was no difference between groups when the maximum tolerable amplitude was normalized to a level required to produce a motor response [40]. A similar outcome was observed with the 30 Hz stimulation protocol: the maximum tolerable amplitude was increased in both groups [40]. The authors suggested that motor threshold may be higher when a carrier frequency is applied because the waveform is suboptimal for spinally evoked muscle response [40]. While the application of a carrier frequency may decrease participants’ discomfort at a given amplitude, it may be offset by an increase in the motor threshold, requiring an increased stimulation amplitude to attain the same therapeutic results. Another observation is that to obtain a desired motor outcome, the use of a carrier frequency was not better than when no carrier frequency was applied [40].

Benavides et al. also provided a great deal of information regarding the utility of carrier frequencies in stimulation protocols [7]. This study tested the effects of TSS with and without a 5 kHz carrier frequency on MEP and short-interval cortical inhibition (SICI) in the biceps brachii. When TSS was applied with a carrier frequency, SICI increased in the carrier-frequency group compared to the non-carrier-frequency group [7]. The use of a carrier frequency may contribute to cortical inhibitory effects, possibly reducing the incidence of spasticity after SCI. Hand and arm function measured by the GRASP test also improved to a greater degree when carrier frequency was used. However, this study only evaluated motor outcomes at a maximum of 75 min following stimulation. It is still unclear how a carrier frequency may affect long-term motor outcomes or enhance neuroplasticity in persons with SCI.

Further exploration is warranted to characterize the role of the carrier frequency properly. To further clarify the role of carrier frequencies in motor outcomes, future studies should emulate Benavides’ and Manson’s designs by including a control group that does not use a carrier frequency.

## 3. Summary/Conclusions

In summary, TSS facilitates improved upper- and lower-extremity function in individuals with SCI. The anatomical positioning of the cathodes either for the lower- or upper-limb programs varied based on the protocol utilized in these studies. Furthermore, multisite stimulation provided better motor outcomes than single-site stimulation. Stimulus amplitude was also highly variable; however, overall, the data supported the use of an amplitude at 90% of the motor threshold to maximize therapeutic benefit. The most selected stimulation frequencies suggested that tonic stimulation that ranged between 30 Hz and 50 Hz is considered the most optimal for either upper- or lower-extremity programs. The use of a carrier frequency was not consistent across all studies. Two studies had conflicting results. One suggested no benefit, and the other study suggested that a carrier frequency may facilitate the recovery of upper-extremity dexterity and enhance cortical inhibitory effects. Therefore, the effects of a carrier frequency on maximizing TSS outcomes remain inconclusive. Identifying the optimal parameters for TSS, based on the available data, is difficult. Future studies are warranted to systematically investigate the effects of manipulating different stimulation parameters to enhance the utilization of TSS in enhancing motor recovery after SCI.

## Figures and Tables

**Table 1 jcm-12-00854-t001:** Electrode characteristics for applications of TSS for motor control in the upper extremities.

(a) Electrode characteristics for applications of TSS for motor control in the upper extremities						
First Author	Year	Demographics	Cathode Size	Cathode Location	Anode Location	Channel Number
Inanici [9]	2018	SCI; ASIA D, C3	2.5 cm diameter	One midline at C3–C4 spinous processes; one midline at C6–C7 spinous processes	Bilateral iliac crests	2
Freyvert [8]	2018	SCI; ASIA B, C5 and higher	Not reported	Dorsal neck overlying C5 vertebrae	Anterior superior iliac spine bilaterally	1
Benavides [7]	2020	SCI; ASIA A–D, C4–C6	3.2 cm diameter	Midline C5–C6 between spinous processes	Bilateral iliac crests	1
Murray [25]	2017	SCI; ASIA C, C6–7	10.2 × 5.1 cm	Midline overlying C5–T2 spinous processes	Bilateral clavicles	1
Inanici [26]	2021	SCI; ASIA B–D, C3–C5	2.5 cm diameter	Midline above and below injury level	Bilateral iliac crests	2
Gad [12]	2018	SCI; ASIA B–C, C7 and higher	2.0 cm diameter	One midline between C3–C4 spinous processes; one midline between C6–C7 spinous processes	Bilateral iliac crests	2
Wu [15]	2020	AB and SCI; C2–C8	5 × 10 cm	Midline 4 cm caudal to C7 spinous process, arranged longitudinally	Horizontally over anterior midline with caudal edge 2–3 cm above sternal notch	1
Kumru [16]	2021	AB	2.0 cm diameter	One midline over spinous processes C3–C4; one midline over spinous processes C6–C7	Bilateral iliac crests	2
Parhizi [21]	2021	AB	2.5 cm diameter	One midline over C3–C4 spinous processes; one midline over C6–C7 spinous processes; one midline over T11 spinous process; one midline over L1 spinous processes	Bilateral iliac crests	4
Sasaki [27]	2021	AB	0.5 × 0.5 cm	Midline over C6 or C7 or T1 spinous processes	Midline on anterior neck	1
de Freitas [28]	2021	AB	5.0 × 5.0 cm	Cathode experiment: over spinous process of C6 vs. C7 vs. T1. Anode experiment: placed at optimum location from cathode experiment	Cathode experiment: midline over anterior neck. Anode experiment: one anode on anterior neck vs. two anodes bilaterally over distal clavicles vs. two anodes bilaterally over iliac crests vs. one anode 4 cm below cathode on posterior neck.	1
Milosevic [18]	2018	AB	5 × 5 cm	Midline between C7–T1 spinous processes	Anterior midline neck	1
**(b) Electrode characteristics for applications of TSS for motor control in the lower extremity**						
**First Author**	**Year**	**Demographics**	**Cathode Size**	**Cathode Location**	**Anode Location**	**Channel Number**
Gorodnicheva [29]	2012	AB	2.5 cm diameter	Midline between spinous processes T11 and T12	Bilateral iliac crests	1
Hofstoetter [10]	2013	SCI; ASIA D, T9	8 × 13 cm	T11/T12 spinous process	Bilaterally over the lower anterior abdomen.	1
Krenn [30]	2013	AB	3 × 12 cm	8 cm caudal and 4 cm rostral around the interspinous space T11–12	Bilateral abdomen	7
Hofstoetter [31]	2014	SCI; ASIA D, C5–T9	5 cm diameter	T11 and T12 spinous processes	Bilaterally over the lower anterior abdomen in symmetry to the umbilicus	1
Bedi [32]	2015	SCI; ASIA C, L1	4.5 × 9 cm	T10–L1 vertebral level	Not reported	1
Sutor [33]	2022	SCI; ASIA A–C, C4–T11	10.2 × 17.8 cm	T10/T11 to L4/L5	Bilateral iliac crests	1
Sayenko [34]	2015	AB	10 mm diameter	Midline spinous processes T10 and L1	Bilateral iliac crests	1 and 2
Gerasimenko [35]	2015	AB	2.5 cm diameter	Midline at C5, T11, and/or L1spinous processes	Bilateral iliac crests	3
Sayenko [36]	2015	AB	18 mm diameter	Between the spinous processes of T10 –T11, T11–T12, and T12–L1 midline	Bilateral iliac crests	3
Gerasimenko [37]	2015	SCI; ASIA A–B	2.5 cm diameter	Midline between spinous processes T11–T12 or over coccyx	Bilateral iliac crests	2
Minassian [13]	2016	SCI; ASIA A	8 × 13 cm	T11 and T12 spinous processes	Covering the abdomen	1
Bedi [38]	2016	SCI; ASIA C, T12–L1	4.5 × 9 cm	T10–L1 para-vertebral	Not reported	1
Shapkova [14]	2020	SCI; ASIA A–C, C5–L2	3 x4 cm	Over T12 vertebra	Centrally over abdomen	1
McHugh [39]	2020	SCI; ASIA C–D, C4–T9	5 × 10 cm	Between T11–T12 spinous process	Over lower abdomen	1
Al’joboori [17]	2020	SCI; ASIA A–D, C5–T10	5 × 5 cm	T10/T11	Over T12/L1	1
Manson [40]	2020	AB	32 mm diameter	Parallel to the spinous process of L1–L2 vertebrae	Over lower abdomen	1
Sayenko [19]	2019	SCI; ASIA A–C, C4–T12	3.2 cm diameter	Between spinous process of T11/T12 and L1/L2	Bilateral iliac crests	2
Gad [11]	2017	SCI; ASIA A, T9–L1	2.5 cm diameter T11/T12, 5.0 × 10.2 cm rectangle pair at Co1	T11–T12 midline between spinous processes T11–T12 (Simply T11) or over Co1	Bilateral iliac crests	2
Gerasimenko [41]	2018	AB	2.5 cm diameter	Between the spinous processes of T11–T12 or L1–L2	Bilateral iliac crests	1
Hofstoetter [42]	2015	SCI; ASIA D, C5–T9	5 cm diameter	T11/T12 paraspinally	Paraumblically	1
Samejima [43]	2022	SCI; ASIA D, C4–C6	2.5 cm diameter	Over midline at C3/C4, C6/C7, T11, and L1	Bilateral iliac crests	2
Bye [44]	2022	SCI; T1–T11	5 × 10 cm	L1/L2	Over lower abdomen	1

**Table 2 jcm-12-00854-t002:** Stimulation amplitude for TSS applications with variable motor thresholds.

Extremity	Threshold Level	First Author	Year	Amplitude Determination	Amplitude
Upper Limb	Submotor threshold	Murray [25]	2017	Below motor threshold to level that induced bilateral muscle contraction	68 mA
		Wu [15]	2020	80–200% of resting motor Threshold	102 mA (80% of the motor threshold)
		Kumru [16]	2021	at 80%, 90%, and 110% of RMT of adductor pollicis brevis	90 mA (80% of the motor threshold)
		Sasaki [27]	2021	Minimum to induce paresthesia	28 mA
		Inanici [26]	2021	To best facilitate each activity	120 mA
	Motor threshold	Freyvert [8]	2018	To maximize voluntary hand contraction	100 mA
		Gad [12]	2018	To maximize grip strength	250 mA
		Milosevic [18]	2018	To evoke responses on ascending portion of recruitment curve of all muscles tested	90 mA
		Benavides [7]	2020	To evoke motor output in biceps brachii	90 mA
		Murray [25]	2017	Below motor threshold to level that induced bilateral muscle contraction	68 mA
	Supramotor threshold	Wu [15]	2020	80–200% of resting motor Threshold	102 mA (up to 200% of the motor threshold)
		Kumru [16]	2021	At 80%, 90%, and 110% of RMT of adductor pollicis brevis	90 mA (110% of the motor threshold)
	Non-specific	Parhizi [21]	2021	At tolerance capacity	70 mA
		Inanici [9]	2018	Unspecified	120 mA
		de Freitas [28]	2021	Cathode experiment: 10–100 mA or at pain threshold; anode experiment: to best produce post-activation depression.	100 mA
Lower Limb	Submotor threshold	Hofstoetter [10]	2013	To produce paresthesia below motor threshold	18 V
		Hofstoetter [31]	2014	To produce paresthesia below motor threshold	22 V
		Bedi [32]	2015	To induce sensory sensation	Unspecified
		Sayenko [34]	2015	10–50% of maximal response amplitude in the LE musculature	100 mA
		Bedi [38]	2016	To induce sensory sensation	Unspecified
		McHugh [39]	2020	Maximum tolerable amplitude or submotor threshold	80 mA
		Hofstoetter [42]	2015	Subthreshold	27 V
		Shapkova [14]	2020	In 1 Hz and 3Hz group, 1.3–1.4 × motor threshold. In 67 Hz group, below motor threshold.	Unspecified
		Samejima [43]	2022	Below motor threshold	75 mA
	Motor threshold	Gorodnicheva [29]	2012	To evoke steplike movements	100 mA
		Krenn [30]	2013	At tolerance capacity (max 125 mA)	125 mA
		Gerasimenko [35]	2015	Based on sensations felt by the subject and the motor output generated	180 mA
		Gerasimenko [37]	2015	To induce stepping-like movements	180 mA
		Minassian [13]	2016	Lower-limb PRM reflex threshold	170 mA
		Gerasimenko [41]	2018	To generate involuntary rhythmic stepping-like movements without causing discomfort	150 mA
		Sayenko [19]	2019	To maximally facilitate standing	150 mA
		Manson [40]	2020	Maximum tolerable amplitude	Unspecified
		Al’joboori [17]	2020	At tolerance capacity or to produce paresthesia, whichever lower	110 mA
		Sutor [33]	2022	At the lowest amplitude that produced lower-extremity EMG output	Unspecified
		Bye [44]	2022	100% of amplitude to cause PRM reflex	Unspecified
	Supramotor threshold	Shapkova [14]	2020	In 1 Hz and 3Hz group, 1.3–1.4 × motor threshold. In 67 Hz group, below motor threshold.	Unspecified
	Unspecified	Sayenko [36]	2015	At tolerance capacity (max 100 mA)	100 mA
		Gad [11]	2017	To best facilitate locomotor activity	200 mA

**Table 3 jcm-12-00854-t003:** Stimulation frequency pattern in the studies that applied or did not apply carrier frequency.

Extremity	Carrier Frequency	First Author	Year	Carrier Frequency (kHz)	Stimulation Frequency (Hz)
Upper-Extremity Studies	Carrier frequency	Inanici [9]	2018	10	30
		Gad [12]	2018	10	30
		Benavides [7]	2020	Either 5 or 0	30
		Inanici [26]	2021	10	30
		Kumru [16]	2021	10	30
		Parhizi [21]	2021	10	30
		Sasaki [27]	2021	10	30
	no carrier frequency	Murray [25]	2017	N/A	0.2
		Freyvert [8]	2018	N/A	30
		Milosevic [18]	2019	N/A	Single pulse
		Wu [15]	2020	N/A	0.2
		de Freitas [28]	2021	N/A	Two 2 ms pulses separated by 50 ms
Lower-Extremity Studies	Carrier frequency	Gorodnicheva [29]	2012	10	1, 5, 10, 20, 30, 40
		Gerasimenko [35]	2015	10	5
		Bedi [32]	2015	2.5	20
		Gerasimenko [37]	2015	10	30 Hz at T11, 5 Hz at coccyx
		Bedi [38]	2016	2.5	30, 50, 70, 90
		Gerasimenko [41]	2018	5	30 at T11–T12, 0.3 at L1
		Sayenko [19]	2019	10	5, 15, 25, 30
		Manson [40]	2020	5	Single pulse 0.2 Hz, continuous 30 Hz
		Bye [44]	2022	10	20 Hz
		Samejima [43]	2022	10	30 Hz
	no carrier frequency	Krenn [30]	2013	N/A	Unspecified
		Hofstoetter [10]	2013	N/A	30
		Hofstoetter [31]	2014	N/A	50
		Sayenko [34]	2015	N/A	30
		Sayenko [36]	2015	N/A	Unspecified
		Minassian [13]	2016	N/A	30
		Gad [11]	2017	N/A	T11: 30 Hz; coccyx segment: 5 Hz
		Shapkova [14]	2020	N/A	1, 3, 67
		McHugh [39]	2020	N/A	50
		Al’joboori [17]	2020	N/A	30
		Sutor [33]	2022	N/A	30
		Hofstoetter [42]	2015	N/A	30

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
