# Peer review of "Optimization of Transspinal Stimulation Applications for Motor Recovery after Spinal Cord Injury: Scoping Review"

_jcm, 2023, doi:10.3390/jcm12030854_

Round 1
Reviewer 1 Report
Overall I find the review very useful and interesting and I think that it fills a gap in the literature.
I have minor comments.
- It might have been useful to write a bit more specifically about the various conditions under which motor tasks - used in outcome measures - were performed in various studies.
Such motor tasks are mentioned as standing, walking, and hand and arm functions.
- For instance, it would further improve the paper if the authors could write about the dependence of the effect of TSS on body position in which motor tasks were performed. E.g. in the case of upper limb movements in sitting or supine or standing positions, the effect of TSS on muscle activities may be different. If there is any information about this in the reviewed papers or in other publications than it would be interesting to include one paragraph about it.
In my view, especially the tables in the paper are comprehensive and informative.
Author Response
Reviewer 1
(x) I would not like to sign my review report
( ) I would like to sign my review report
English language and style
( ) English very difficult to understand/incomprehensible
( ) Extensive editing of English language and style required
( ) Moderate English changes required
( ) English language and style are fine/minor spell check required
(x) I don't feel qualified to judge about the English language and style
|
Is the work a significant contribution to the field? |
|
|
Is the work well organized and comprehensively described? |
|
|
Is the work scientifically sound and not misleading? |
|
|
Are there appropriate and adequate references to related and previous work? |
|
|
Is the English used correct and readable? |
Comments and Suggestions for Authors
Overall I find the review very useful and interesting and I think that it fills a gap in the literature. I have minor comments.
It might have been useful to write a bit more specifically about the various conditions under which motor tasks - used in outcome measures - were performed in various studies. Such motor tasks are mentioned as standing, walking, and hand and arm functions.
Authors: The specific anatomical positioning for motor outcomes has been, addressed.
For instance, it would further improve the paper if the authors could write about the dependence of the effect of TSS on body position in which motor tasks were performed. E.g. in the case of upper limb movements in sitting or supine or standing positions, the effect of TSS on muscle activities may be different. If there is any information about this in the reviewed papers or in other publications than it would be interesting to include one paragraph about it.
Authors: Current literature has not addressed this point, it is quite interesting and has been added in the summary paragraph in the electrode placement section.
In my view, especially the tables in the paper are comprehensive and informative.
Authors: We would like to thank the reviewer for his time and effort and his important feedback about our manuscript.
Reviewer 2 Report
1.The content is novel and has clinical significance.
2.The introduction part is too tedious, so it can be properly simplified and the key information can be explained clearly.
3.The table is only a summary of information, but not through processing and integration, it need to be modified.
4.It seems that no valid conclusion has been reached in the conclusion part, and further information mining is needed to draw a conclusion.
5.Figures can be used to illustrate more intuitively.
Author Response
Reviewer 2
Open Review
(x) I would not like to sign my review report
( ) I would like to sign my review report
English language and style
( ) English very difficult to understand/incomprehensible
( ) Extensive editing of English language and style required
( ) Moderate English changes required
( ) English language and style are fine/minor spell check required
(x) I don't feel qualified to judge about the English language and style
|
Is the work a significant contribution to the field? |
|
|
Is the work well organized and comprehensively described? |
|
|
Is the work scientifically sound and not misleading? |
|
|
Are there appropriate and adequate references to related and previous work? |
|
|
Is the English used correct and readable? |
Comments and Suggestions for Authors
1.The content is novel and has clinical significance.
Authors: We would like to thank the reviewer for his time and effort and his important feedback about our manuscript.
2.The introduction part is too tedious, so it can be properly simplified and the key information can be explained clearly.
Authors: Per Reviewer request, the sections in the introduction have been added to establish the effectiveness of TSS; however, effort has been made to reduce the wordiness of the introduction and to make it more specific.
3.The table is only a summary of information, but not through processing and integration, it needs to be modified.
Authors: The comment is really appreciate it, a more specific comment on what should be addressed would be more helpful, as one of the reviewer has stated that the tables are “comprehensive and informative”
- It seems that no valid conclusion has been reached in the conclusion part, and further information mining is needed to draw a conclusion.
Authors: The review tries to summarize the protocols that are being utilized. A more specific conclusion has been added in the summary section with more definitive outcome.
5.Figures can be used to illustrate more intuitively.
Authors: Tables were chosen instead of figures to summarize the information in a more concise manner.
Reviewer 3 Report
Please provide a better short explanatory title for table 2.
Did authors provide a PRISMA checklist in their submission?
The authors stated in the abstract (line 24 ) : TSS protocols and their effects on limbs function were summarized however, after comparing the electrode placement in studies based on the targeted limbs, no overall functional effects is reported. To improve the relevance of the review to the clinical application of TSS, I suggest to find a few functional outcomes that were reported with most reported studies and try to bring a more comprehensive picture of the used protocols and their resulting effects on function. You can include the ASIA scores (where available), sensory and motor impairments in the included SCI population for each study and then summarize the common stimuli protocols resulted in motor/function/sensory gains.
The JCM journal is not limiting authors to follow a particular format for papers however to increase clarity, it is recommended that authors separate the results from the methods and discussion sections.
Please include citations for studies on the TSS-enables functions (line 120).
Line 248. table 1a. last column de Freitas 1551 ? is this a typo?
After reading the extensive reports on the protocols utilized on able-bodies adults, I am still unsure its relevance to this scoping review. Please reconsider its relevance and clarify.
Author Response
Reviewer 3
Open Review
(x) I would not like to sign my review report
( ) I would like to sign my review report
English language and style
( ) English very difficult to understand/incomprehensible
( ) Extensive editing of English language and style required
( ) Moderate English changes required
( ) English language and style are fine/minor spell check required
(x) I don't feel qualified to judge about the English language and style
|
Is the work a significant contribution to the field? |
|
|
Is the work well organized and comprehensively described? |
|
|
Is the work scientifically sound and not misleading? |
|
|
Are there appropriate and adequate references to related and previous work? |
|
|
Is the English used correct and readable? |
Comments and Suggestions for Authors
Please provide a better short explanatory title for table 2.
Authors: The title of the table has been rephrased.
Did authors provide a PRISMA checklist in their submission?
Authors: This is a scoping review thus a Prisma checklist would not be suitable
The authors stated in the abstract (line 24 ) : TSS protocols and their effects on limbs function were summarized however, after comparing the electrode placement in studies based on the targeted limbs, no overall functional effects is reported. To improve the relevance of the review to the clinical application of TSS, I suggest to find a few functional outcomes that were reported with most reported studies and try to bring a more comprehensive picture of the used protocols and their resulting effects on function. You can include the ASIA scores (where available), sensory and motor impairments in the included SCI population for each study and then summarize the common stimuli protocols resulted in motor/function/sensory gains.
Authors: The comment is really appreciated, motor outcome based on parameters has been discussed in the sections.
Line 216 – line 222, highlights the motor outcome in the section regarding electrode placement.
Line 309-311, Line 329- line 334 highlights the motor outcome in section regarding stimulus amplitude.
The JCM journal is not limiting authors to follow a particular format for papers however to increase clarity, it is recommended that authors separate the results from the methods and discussion sections.
Authors: We really appreciate your point. However, this is a scoping review and not study to separate the sections. We are trying to address each point in scientific sequence to the importance of the work.
Please include citations for studies on the TSS-enables functions (line 120).
Authors: Citation has been added.
Line 248. table 1a. last column de Freitas 1551 ? is this a typo?
Authors: The typo has been corrected.
After reading the extensive reports on the protocols utilized on able-bodies adults, I am still unsure its relevance to this scoping review. Please reconsider its relevance and clarify.
Authors: The aim of the study was to highlight the protocols that have been utilized by other research groups as far as electrode placements, stimulation parameters and different waveforms. The inclusion of able-bodied studies provides the reader with clear picture of what has been accomplished and how relevant protocols were modified in managing upper and lower extremity motor controls in persons with SCI. They also provide insight to the mechanism behind TSS.
Round 2
Reviewer 3 Report
The new changes has increased the clarity of the manuscript.
Please ensure to include a new column in tables 1a&b to show population demographics (healthy or injured, and if injured type and the level of injury ASIA scores, etc) for the included studies. This would increase the clinical relevance of this scoping review.
line 441 In table 2 did Wu and Kumru used the same amplitude determination protocols? The information for these two rows in the table are messed up right now both for sub and supra motor TS.
Author Response
The new changes has increased the clarity of the manuscript.
Authors: Thank you so much for taking the time to review our manuscript and provide highly important feedback.
Please ensure to include a new column in tables 1a&b to show population demographics (healthy or injured, and if injured type and the level of injury ASIA scores, etc) for the included studies. This would increase the clinical relevance of this scoping review.
Authors: We have now included this information based on your request to both tables 1a and 1b.
line 441 In table 2 did Wu and Kumru used the same amplitude determination protocols? The information for these two rows in the table are messed up right now both for sub and supra motor TS.
Authors: Both Wu and Kumru et al. used the same protocol for determination of motor threshold. We have chosen to list them twice because they started their TSS protocol at sub-motor threshold (Wu et al. started 80% of motor threshold and Kumru et al. started at 80%) and then both progressed to supra-motor threshold (Wu et al progressed to 200% of motor threshold without a carrier frequency, whereas kumru et al. progressed to 120% with a 10 kHz carrier frequency).